# Use of a modular ontology and a semantic annotation tool to describe the care pathway of patients with amyotrophic lateral sclerosis in a coordination network

**Sonia Cardoso**[1], **Pierre Meneton**[1], **Xavier Aimé**[1,2], **Vincent Meininger**[3], **David Grabli**[4], **Gilles Guezennec**[1], **Jean Charlet**[1,5]*

**1** Laboratoire d'Informatique Médicale et d'Ingénierie des Connaissances en e-Santé UMR-1142, Sorbonne Université, INSERM, Université Paris 13, Paris, France, **2** Cogsonomy, Nantes, France, **3** Ramsay General de Santé, Hôpital Peupliers, Paris, France, **4** Département des maladies du Système Nerveux, Assistance Publique-Hôpitaux de Paris Pitié Salpêtrière, Paris, France, **5** Département de la Recherche Clinique et de l'Innovation, Assistance Publique-Hôpitaux de Paris, Paris, France

* jean.charlet@sorbonne-universite.fr

**Data Availability Statement:** All relevant data are within the manuscript and its Supporting information files.

## Abstract

The objective of this study was to describe the care pathway of patients with amyotrophic lateral sclerosis (ALS) based on real-life textual data from a regional coordination network, the Ile-de-France ALS network. This coordination network provides care for 92% of patients diagnosed with ALS living in Ile-de-France. We developed a modular ontology (Onto-PaRON) for the automatic processing of these unstructured textual data. OntoPaRON has different modules: the core, medical, socio-environmental, coordination, and consolidation modules. Our approach was unique in its creation of fully defined concepts at different levels of the modular ontology to address specific topics relating to healthcare trajectories. We also created a semantic annotation tool specific to the French language and the specificities of our corpus, the Ontology-Based Semantic Annotation Module (OnBaSAM), using the OntoPaRON ontology as a reference. We used these tools to annotate the records of 928 patients automatically. The semantic (qualitative) annotations of the concepts were transformed into quantitative data. By using these pipelines we were able to transform unstructured textual data into structured quantitative data. Based on data processing, semantic annotations, sociodemographic data for the patient and clinical variables, we found that the need and demand for human and technical assistance depend on the initial form of the disease, the motor state, and the patient age. The presence of exhaustion in care management, is related to the patient's motor and cognitive state.

## Introduction

Neurodegenerative diseases affect many people in France, and throughout the world. For example, Parkinson's disease affects about 160,000 people in France [1] and nearly one million in the United States https://www.parkinson.org/Understanding-Parkinsons/Statistics. Alzheimer's disease and related conditions affect more than one million people in France, and it was

**Funding:** Xavier Aimé works in the company Cogsonomy that he created: This company's sole role is to finance training and other project management assistance for Xavier Aimé. Cogsonomy did not play a role in the study design, data collection and analysis, decision to publish, preparation of the manuscript and only provided financial support in the form of authors' salaries and/or research materials. The funders (universities, hospitals) provided support in the form of salaries for authors [SC, PM, VM, DG, GG, JC], but did not have any additional role in the study design, data collection and analysis, decision to publish, or preparation of the manuscript. The specific roles of these authors are articulated in the 'author contributions' section.

**Competing interests:** Xavier Aimé works in the company Cogsonomy that he created. This commercial affiliation (Cogsonomy) does not alter our (all the authors) adherence to PLOS ONE policies on sharing data and materials.

estimated that 5 million Americans were living with this disease in 2014 https://www.cdc.gov/aging/aginginfo/alzheimers.htm. In Europe, the incidence of amyotrophic lateral sclerosis (ALS) has been estimated at 2.2 per 100,000 person-years (py) for the general population [2]. In 2015, 16,583 people were identified as having ALS in the USA [3]. In France, the annual incidence of ALS has been estimated at 1,500 new cases [4]. These diseases have many features in common: they cause a number of disabilities and handicaps that cannot be cured, although the symptoms can be treated with pharmacological and non-pharmacological approaches. Neurodegenerative diseases cause polymorphic damage (motor impairment, respiratory impairment, cognitive impairment, etc.), requiring the intervention of multiple structures and professionals. In France, the professionals and structures involved in the care pathway come from three specific sectors (medical, social, and medico-social). The care pathway includes support provided at home, but also passages through hospitals or care structures.

In France, care pathway optimization and the maintenance of care continuity have become key health policy issues in recent years. Care pathway optimization should improve patient management and have a positive economic impact on the healthcare system by limiting hospitalizations (particularly those that are avoidable) and unnecessary medical procedures. However, our knowledge of care pathways and their components, including their interruptions and the difficulties encountered by patients and their families, remains partial, because information is split between various actors and is often untraceable. The identification of interruptions of patient care requires a prior knowledge of the elements of the care pathway. We addressed these issues, using a real-life database for patients with ALS managed by the regional coordination network of Ile-de-France.

ALS is a neurodegenerative disease that affects the motor neurons, causing progressive weakening of the voluntary muscles. This disease has two forms: a familial form, accounting for 10% of cases, and a sporadic form accounting for the other 90%. Median survival after diagnosis is generally three years [5]. Progressive paralysis of the muscles generates functional limitations, causing disability (loss of the ability to walk, difficulty speaking, loss of dexterity), and ultimately resulting in death. The disease has an impact not only on the patient, but also on the patient's family and careers [6]. Over and above management of the medical aspects of the disease, the patients and their families need compensation and assistance. This assistance may take the form of (a) human help with activities of daily living (eating, washing, getting dressed, etc.) or (b) technical assistance for mobility or communication, such as a powered wheelchair, or a speech synthesizer [7, 8]. This human and technical assistance is costly, and patients and their families require social assistance to complete the necessary administrative procedures to attain such funding [9].

In France, ALS patients are managed partly by expert centers. A specific group sector of such centers has been established: Rare Diseases, Amyotrophic Lateral Sclerosis, and Motor Neuron Diseases (https://portail-sla.fr). In Paris, a regional coordination network was created in 2005: the Ile-de-France ALS network (ALS-IDF network). The objective of this network is to coordinate actions and to help patients, families, and professionals through holistic management encompassing medical, social and medico-social aspects during the various stages of disease progression. A studied published in 2015 [10] reported that 92% of patients with ALS in Ile-de-France were managed by the ALS-IDF network. This study revealed an impact of this coordination on the number of hospital admissions and an improvement in survival. The ALS-IDF network established a database to make it easier to track the requests, needs and coordination actions implemented to support patients. This database contains two types of patient data: real-life and sociodemographic data.

We hypothesized that an analysis of these textual data would make it possible to identify the difficulties and needs of patients and their families at home, to understand the coordination

actions implemented and to identify situations or types of patients confronted with multiple difficulties. The identification of such situations should improve patient support and the early detection of risk situations. We processed these textual data by semantic annotation using an ontology corresponding to this domain as a lexical resource. An ontology is defined as the formalization of a shared conceptualization [11]. Ontologies, as conceptual models, provide the necessary framework for semantic representation of textual information. Several ontologies have been developed in neurology, for Alzheimer's disease [12], the Parkinson's disease [13], and neurological diseases in general [14]. In this context, the work done on Alzheimer's ontology [12] is inspiring for us since the authors have developed an ontology with French and English terms in order to annotate an information portal. However, none of the existing ontologies simultaneously models all of the knowledge relating to ALS, care pathway coordination and the specific features of existing social and medical structures in France.

We have therefore developed our own ontology, OntoPaRON, including these different dimensions in a modular structure. A modular ontology corresponds to a set of modules, where each module is a stand-alone component that maintains relationships with other ontology modules [15]. A modular ontology seemed be the most appropriate model for taking all these aspects into account. As a means of focusing our research on specific themes and identifying the difficulties encountered by patients during their care pathways, we decided to create fully defined concepts [16], each encompassing several classes relating to the same theme but from different modules.

We developed a semantic annotation tool based on the General Architecture for Text Engineering (GATE https://gate.ac.uk) open framework, which provides basic building blocks for the annotation of textual data. GATE's resources have been adapted to our needs and to the French language. Our annotator, Ontology-Based Semantic Annotation Module (OnBaSAM) uses the OntoPaRON ontology. We used the annotations performed to create a specific module with annotation frequency as output. One of the advantages of this work is that it makes it possible to convert unstructured textual data into structured quantitative data. These quantitative data can be used for population-based statistical approaches in which the annotation data obtained are used to describe the elements of patient care trajectories. The work includes information extraction using ontology and specifically fully defined concepts, and data mining tasks by looking for relationships between the information extracted by semantic annotation and the socio-demographic data from the ALS-IDF network database [17].

## Materials and methods

Computer tools are essential for exploitation of the textual data of the ALS-IDF network, which include too great a volume of information for manually processing. We decided to use knowledge engineering and automatic natural language processing (NLP) tools. We present here: (a) the use of an ontology to model domain knowledge; (b) the creation of a fully defined concept concerning specific themes relating to the care pathway; (c) the use of a semantic annotation tool that we developed, the OnBaSAM module. All these analyses were performed with JMP 14 Pro Statistical Discovery software (SAS, Cary NC).

### Materials

The ALS-IDF network database has two parts: a structured static part containing sociodemographic data for the patients (sex, date of diagnosis, date of inclusion in the network, living conditions, date of birth, place of residence, etc.), and a dynamic part consisting of real-life data in a textual format. Textual data for "events" (unstructured part of the database) are entered as free text by the coordinators and may be of different types (e.g. transmission,

hospitalization reports, minutes of care team meetings, medical transcriptions, or the transcription of oral exchanges with patients or their families). The Table 1 presents two examples of events present in the database. The processing of textual event data required initial spelling correction and pseudo-anonymization to comply with the General Data Protection Regulation (GRPD) rules. On August 26, 2019, the database of the ALS-IDF network contained 2,684 patient files, including more than 80,000 textual entries for events.

## Construction of the OntoPaRON ontology

We use 'single quotes' to denote OntoPaRON classes and *italic font* to denote the relationships in our ontology. The specific features of our project relate to (a) the decision to model knowledge through a modular ontology and (b) the use of fully defined concepts specifically created as themes of interest in statistical analyses, to improve clinical understanding of the care trajectory. Our ontology was developed with a methodology combining a top-down approach involving the use of a top-level ontology with a bottom-up approach involving searches for candidate terms in the corpus of the text [18].

The first step was the extraction of corpus terms with NLP tools in BIOTEX software [19]. The corpus used consisted of 60,130 events extracted from the database of the ALS-IDF network, covering a ten-year period of network activity (2005 to 2015). We selected the candidate terms for this analysis in collaboration with experts in the field. This method provides access to terms representing the concepts used. The OntoPaRON ontology was constructed with Protégé (https://protege.stanford.edu) version 5.2 [20]. Each concept in our ontology is denoted by a preferred term in English and in French, together with alternative terms (synonyms, acronyms, and abbreviations taking into account different spellings linked to the coordination context). Indeed, there is no consensus between the coordinators of the ALS-IDF network (who come from different paramedical professions: nurse, occupational therapist, psychologist) on the common use of abbreviations. This diversity of usage required the collection of all the terms and abbreviations used for the same concept. For example, the concept 'general practitioner' may be denoted by eight alternative terms in french: 'med tt', 'mt', 'family doctor', 'med ttt', 'méd t', 'méd tt', 'general practitioner', 'mdt'.

**Table 1. Examples of events recorded in the SLA-IDF network database.**

| In FRENCH | In ENGLISH |
|---|---|
| Information du ssiad de VILLE, PROFESSIONEL: a débuté la PEC ce jour mais le logement n'est pas du tout adapté. Nous ne pourrons pas lui faire la douche car la SDB n'est pas du tout adaptée. Son épouse prend beaucoup de risques dans les transferts le patient pèse 103 kg il y a un risque de chute non négligeable. | Information from the care at home service of the patient's town of residence, message from a PROFESSIONAL: management initiated today but the accommodation is entirely unsuitable. We will not be able to help him to shower because the bathroom is completely unsuitable. The patient's wife takes a lot of risks when transferring the patient: the patient weighs 103 kg and there is a non-negligible risk of falls. |
| Appel de Patient qui a sollicité COORDINATEUR SLA, pour obtenir un certificat médical du NEUROLOGUE pour son dossier Mdph. Demande si COORDINATEUR SLA peut le rappeler pour son pb de FRE. | Call from a patient who has requested an ALS COORDINATOR, to obtain a medical certificate from the NEUROLOGIST for his local disability services file. Asked if ALS COORDINATOR can call him back for his electric wheelchair problem. |

These examples highlight the frequent use of abbreviations by coordinators (for example, PEC for patient management, SDB for bathroom, FRE for electric wheelchair). Pseudo-anonymization led to the replacement of names with functions, facilitating the annotation. The transformation of a nominal data Dr Brain, not defined in the ontology, into data identifying a concept the agent 'Neurologist' defined in the ontology allows from a conceptual point of view to identify the interactions between agents and actions.

The second step was the alignment of concepts and enrichment of the ontology. For this step, we used the Health Terminology Ontology Portal (HeTOP (https://www.hetop.eu/hetop/)) [21] tool to align the concepts of our ontological modules with other reference terminologies, using Unified Medical Language System (UMLS) codes.

The analysis of candidate terms led to the definition of four principal dimensions:

- a) a generic dimension, corresponding to the set of concepts common to all themes present in the care pathway;

- b) a medical dimension associated with the disease and medical management processes;

- c) a socio-environmental dimension linked to the social situation and environment of the patient;

- d) a coordination dimension linked to the actions of ALS-IDF network coordinators to help patients with administrative formalities, such as finding appropriate healthcare professionals and evaluating needs.

These four dimensions oriented us towards the creation of four ontological modules for each of these domains.

## Modularity of OntoPaRON

We chose to construct a modular ontology, consisting of four domain modules and one a consolidation module. The modules are autonomous but have a defined association with other ontology modules, including the original ontology [22]. Modularity has several advantages, including module reuse and the facilitation of management, by module [23, 24]. We chose to create a modular ontology for several reasons: the possible secondary use of some of the modules for care trajectory analysis for other neurodegenerative diseases (e.g. Alzheimer's disease, Parkinson's disease), facilitation of the updating and handling of knowledge by taking the evolution of systems into account (social assistance system, medical advances), and the promotion of exchanges with experts in the field (doctors, coordinators). The OntoPaRON ontology is composed of five modules (one core module, three domain modules, one consolidation module):

1. The core module, which contains all the high-level concepts common to the three ontologies, such as 'ideal objects','agents', 'processes', and 'modes'. Some of the core ontology and high-level concepts are shown in Fig 1. This ontology was inspired by the Menelas top-level ontology [25] available from: https://bioportal.bioontology.org/ontologies/TOP-MENELAS. This module also contains all object properties used, and the fully defined high-level concepts.

2. The medical module is the largest in terms of the number of classes. It is specific to the medical domain and includes: 'medical agents' ('doctor', 'neurologist', 'physiotherapist', etc.), 'medical processes' ('consultation', 'hospitalization', etc.), and 'medical objects' ('drugs', 'prescriptions', etc.). This module contains concepts relating to anatomical structures, signs, and symptoms. These concepts are directly related to the disease and its medical management.

3. The socio-environmental module contains concepts relating to the life of the patients, in their family and social environments [26]. This module includes 'agents' ('family', 'social workers', etc.),'social actions' ('requests for benefits', 'requests for human assistance', etc.), 'physical objects' from the social domain ('wheelchair', 'medical insurance card', 'accommodation', etc.), concepts linked to the various benefits the patient may receive and

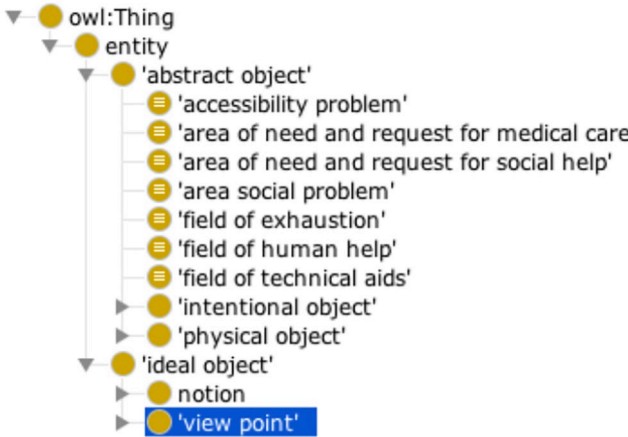

**Fig 1. Core module of the OntoPaRON ontology.** High-level concepts, such as 'abstract objects' and 'ideal objects', are presented in this module, together with concepts defined as high-level.

concepts relating to legal protection ('guardianship', 'ward of court'). The socio-environmental module has the second largest number of classes.

4. The coordination module consists mostly of specific coordination missions ('coordination actions'). There are several types of coordination action: 'communication actions', 'assessment of needs actions', and 'resource search actions'. We based our model partly on a previous study [27] describing the creation of the Nursing Care Coordination Ontology (https://bioportal.bioontology.org/ontologies/NCCO). This is the module with the smallest number of concepts.

5. The last module is a consolidation module. This module does not model any concepts, but all the concepts of the ontology are imported into it. Following importation of the various modules, the HermiT reasoner (1.3.8.413) [28] from Protégé is used to infer classes under the fully defined concept. A reasoned version of the ontology, with the concepts inferred under the fully defined concept, is exported and used by the two tools we developed: the semantic annotator Ontology-Based Semantic Annotation Module (OnBaSAM), and the annotation evaluation tool Pronto.

The importation links between the various modules of OntoPaRON are illustrated in Fig 2, and the metrics for each module of the ontology are shown in Table 2. OntoPaRON is available from https://bioportal.bioontology.org/ontologies/ONTOPARON. The ontology respects a certain number of metadata according to the prescriptions of [29]. To see the classification after reasoning, download the ontology from Bioportal, load it in Protégé and start reasoning with Hermit. The final reclassification is visible in the "class hierarchy (inferred)" view.

## Construction of fully defined concepts

In this case of use, we wished to define the elements of the care trajectory of the patient and to determine whether these elements were expressed in a similar manner for all patients. We hypothesized that semantic annotation of the event database of the ALS-IDF network with our annotation tool, OnBaSAM, using the OntoPaRON ontology as a reference, would help us to understand the care trajectories of patients. We wanted to know whether certain themes, such

http://www.limics.fr/ontologies/ontoparonnoy

http://www.limics.fr/ontologies/ontoparonmed

http://www.limics.fr/ontologies/ontoparoncoord

http://www.limics.fr/ontologies/ontoparonsoc

http://www.limics.fr/ontologies/ontoparon

**Fig 2. Overview of the OntoPaRON concept inheritance diagram.** Overview of the inheritance diagram of the OntoPaRON concept and the corresponding URIs. The arrows point in the direction of the modules that perform the import. Thus, the ontologies of the domain (ontoparonmed: medical ontology, ontoparonsoc: socio-environmental ontology; ontoparoncoord: coordination ontology) import the core ontology. In the same way OntoPaRON ontology imports each of the ontologies of the domain and by inference the core ontology.

as 'exhaustion', 'obtaining human assistance', or 'seeking help to find a healthcare professional', were frequent in patient care trajectories or whether they concerned all patients. Exhaustion during patient management may be expressed in several different ways: the spouse may explicitly report being exhausted, the use of respite care (requested by the patients or their families, or proposed by a healthcare professional), or the exhaustion of teams caring for the patient at home. We created fully defined concepts to bring together all the concepts relating to the same theme.

The organization of knowledge into an ontology made it possible to construct fully defined concepts with sufficient and necessary conditions. Fully defined concepts make it possible to group together all concepts linked to the same theme by the same object property, in the same class. These concepts may be found at different levels in the ontology or in different ontology modules. The HermiT reasoner infers the membership of all concepts sharing a relationship to a fully defined concept defined with this same relationship. The fully defined high-level concepts found in the core module are illustrated in Fig 3. The 'domain of exhaustion' thus includes the exhaustion of a carer from the family or of a professional carer during patient

**Table 2. Metrics of the OntoPaRON ontology modules.**

|  | Core | Medical | Socio-environmental | Coordination | Consolidation | OntoPaRON |
|---|---|---|---|---|---|---|
| Number of classes | 378 | 1041 | 740 | 303 | 0 | 2,462 |
| Number of relationships | 32 | 0 | 0 | 0 | 0 | 32 |
| Fully defined concepts | 7 | 17 | 10 | 9 | 0 | 43 |

The number of fully defined classes, relationships and concepts present in each module of the OntoPaRON ontology. As shown in the table, the consolidation module does not model any concepts; its main function is to aggregate the four modules.

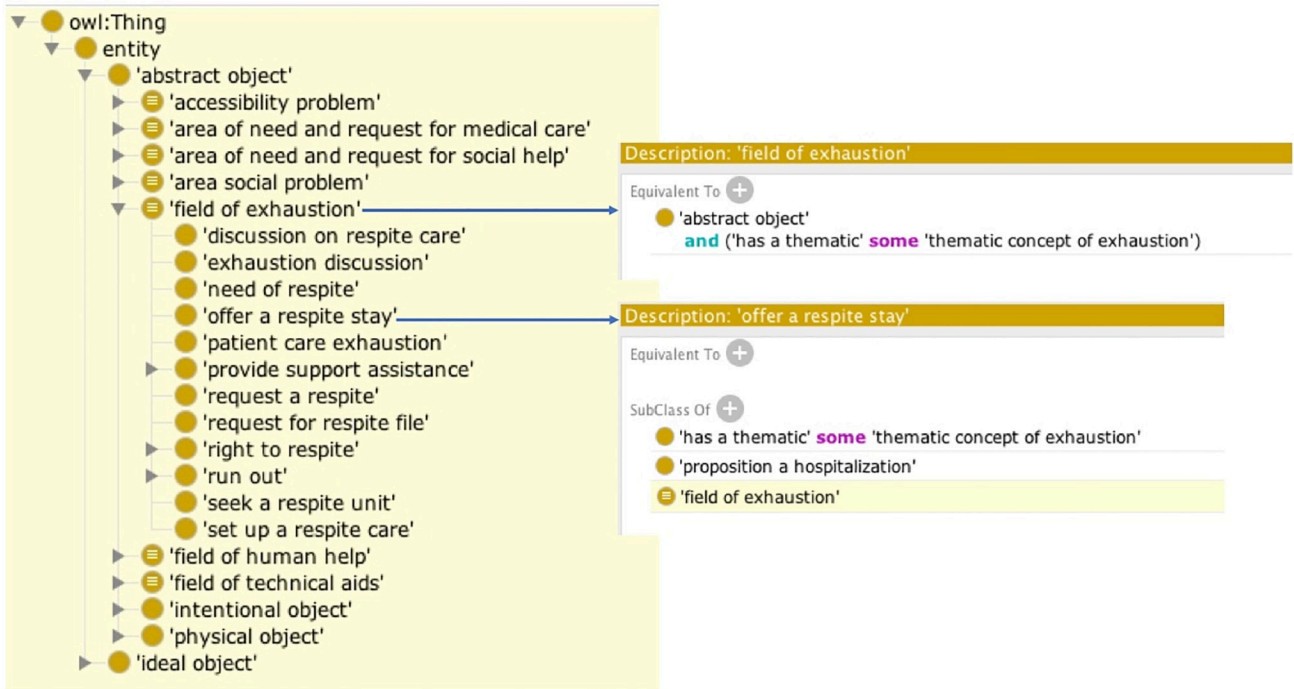

**Fig 3. Fully defined concepts in OntoPaRON.** Partial screenshot of the high-level concepts of OntoPaRON present in the core module. All the concepts present in each of the modules having the relation *thematic concept of exhaustion* are inferred under the fully defined concept 'field of exhaustion'.

management. The concepts relating to this theme are found in two ontology modules: the concepts 'request for respite care' and 'respite care proposal' are found in the coordination module, whereas the concept 'need for respite care' and 'patient carer exhaustion' are present in the socio-environmental module. These three concepts have the object property *aPourThématique ThématiqueEpuisement* and are therefore inferred under the fully defined concept of 'Exhaustion Domain'.

Fully defined concepts are constructed and created as variables of interest for the clinical analysis of patient care trajectory, and their frequency is an indicator of the problems encountered by patients. After annotation of the corpus by the OnBaSAM system, all the related quantitative data linked to the occurrence of exhaustion in caring for a patient can be extracted. Searches for exhaustion of a carer or healthcare professional are generally performed according to a methodology based on individual interviews or rating scales [30–32]. Our approach makes it possible to determine the frequency of this phenomenon in the management of ALS patients from textual data entered by third parties.

Fig 3 illustrates the fully defined concepts presented in the core module, but others are present in each of the modules. The fully defined concept 'Coordination Action', which groups together all the actions performed by the coordinators, is present in the coordination module. It encompasses all the 'coordination actions', including 'communication actions' and 'finding coordination resources'. As another example, we created the fully defined concept 'Social Process', which brings together all requests and actions in the social domain, including 'request for social benefits' and 'bathroom adaptation' for the social module. In the medical module, we created the fully defined concept 'Cognitive State', which brings together all concepts referring to clinical signs, symptoms, and diagnoses linked to changes in the cognitive state of the patient, and 'Motor state', which reports clinical signs relating to a deterioration of the

patient's motor skills, such as falls or losses of motricity. In total, the ontology includes 43 fully defined concepts, part of defined concepts and formal definitions are presented in S1 Table.

## Level of alignment with other terminological and ontological resources

Our studies of existing terminological/ontological resources (TORs) identified none including the dimensions of the patient care trajectory. This finding justifies our ontological approach. Once the complete ontology is obtained, it can be interesting to revalidate the approach a posteriori and to check whether any known terminological/oncological resources cover the domain modeled. We therefore assessed the level of coverage of each module of the Onto-PaRON ontology by the ontologies present in HeTop (https://www.hetop.eu/hetop/), using the terms in French. The HeTop terminology server provides access to 85 TORs. Automatic unsupervised alignment identified found 9,906 alignments for 51 TORs. The results are summarized in Table 3.

For the three modules, the best alignment rates were those obtained with three terminologies: Mesh, SNOMED-CT, and NCIt. These results are readily explained by the specific modeling choices made during the construction of the ontology. Indeed, the medical module contains all the medical professionals, drugs, signs, and symptoms present as generic elements in many resources, without and not specific to ALS. Thus, certain symptoms, such as headache, or the presence of motor disturbances (e.g. falls or muscle amyotrophy), may also be found in other diseases. The results for the socio-environmental module are readily explained by the identification of generic classes, such as family and carer. However, it is difficult to identify benefits or structures specific to the French domain in the terminologies, for example: a) the *prestation sociale* (social benefits) class was found by only two of the terminologies (Mesh et SNOMED-CT), but, concepts at a finer level of granularity, such as *Allocation Adulte Handicapé* (disabled adult allowance), were not present in any of the terminologies and; b) no alignments were found within the framework of medicosocial structures specific to France. Thus, specific structures, such as the *Maison départementale des personnes handicapées ou bien encore le SAVS (Service d'Accompagnement à la Vie Sociale; social assistance)* were not present in any of the terminologies. The coordination module was the module with the lowest percentage alignment. This is partially explained by the level of granularity defined for the coordination module. Thus, the *Establishment of Coordination Resources* class was aligned with the MeSH, SNOMED-CT, and NCIt terminologies. However, no alignments were found for subclasses, such as the *establishment of a respite care stay* or the *establishment of disability benefits* in any of the TORs. In this case of use, it was important to know the specific arrangements made by the coordinators, which could: a) respond to a request clearly expressed by patients

**Table 3. OntoPaRON alignments with terminological/ontological resources presents in HeTop.**

| Terminology | Number of Concepts | Alignments found | Socio-environmental alignments | Medical alignments | Coordination alignments |
|---|---|---|---|---|---|
| Medical subject headings (MeSH) | 277,575 | 789 | 181 (27.01%) | 501 (51.6%) | 14 (3.47%) |
| Systematized Nomenclature Of Medicine Clinical Terms (SNOMED-CT) | 350,976 | 763 | 163 (24.32%) | 432 (44.5%) | 14 (3.47%) |
| National Cancer Institute Thesaurus (NCIt) | 79,870 | 698 | 127 (18.95%) | 370 (38.10%) | 15 (3.72%) |
| International Classification of Diseases (ICD-11) | 55,267 | 310 | 39 (5.82%) | 243 (25.02%) | 3 (0.74%) |

Results of alignments by ontological module of OntoPaRON with the TORs presented in HeTop. The table shows for each module of the ontology OntoPaRON the number of concepts aligned with the reference terminology, and the percentage of terms aligned in each module. For the four TORs considered, the alignments for the medical module were of better quality than those for the coordination module ($p < 0.0001$).

or their families or b) result from the analysis of a need not identified by the patient but highlighted by the coordinator. This analysis clearly shows that no other TORs comes close to covering the whole of our domain.

## Annotation and assessment tools: The Ontology-Based Semantic Annotation Module and Pronto annotator

Within the Medical Informatics and eHealth Knowledge Engineering Laboratory (LIMICS), we decided to develop a semantic annotator using OntoPaRON ontology as a semantic reference. We created this annotator OnBaSAM, with resources available from the GATE, a text analysis platform providing open source resources. These resources are used in various biomedical research projects [33]. Using the available resources, we constructed annotation chains for textual documents. Based on GATE which allows to build a semantic annotation string using an ontology. We have built a specific processing chain that we have adapted to our use case. Indeed, GATE is used and developed in English. Based on GATE, we have built a specific processing chain for our problem, which allows us to take into account the French language, the negation, as well as the export of annotations. At the end of an annotation chain, the textual documents of the corpus are enriched by metadata annotations represented by XML tags included in the annotated document. We created various pipelines (chains of processing resources), allowing several levels of corpus processing:

1. Pre-processing by normalization and tokenization, splitting into sentences, application of lemmatization (TreeTagger), and Part Of Speech (POS) tagging, to make use of grammatical categories. This pipeline can be used to correct spelling in the content, thereby favoring the identification of concepts during annotation.

2. The second pipeline can be used to annotate entities named according to the identification of the ontological concepts from the preflLabel and altLabel of the concept.

3. We chose the option of exporting the created annotations to a spreadsheet. For each patient, the number of occurrences identified by OnBaSAM is determined for each ontology class. The export process can be global, taking into account all ontological concepts ($n = 2, 462$), or specific, restricted to defined concepts only ($n = 43$).

We created Pronto, a tool for the evaluation of annotations, for assessments of the quality of the automatic annotations made a by the OnBaSAM system. This tool can be used to group the annotated text and the corresponding concepts together on the same interface, as shown in Fig 4. This tool is designed for use by evaluators who are experts in the field, directly involved in the coordination of patient care trajectories. Internal and external coordinators from the ALS-IDF network assisted with the assessment of annotation quality. During the evaluation process we asked the coordinators both to evaluate the concepts annotated by the system, and

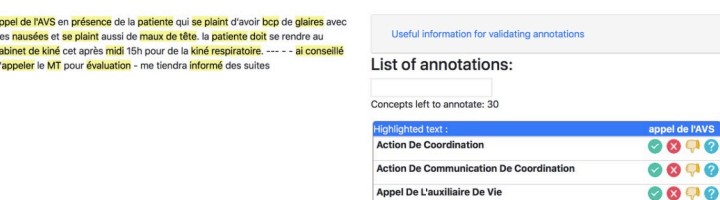

**Fig 4. Pronto assessment tool.** Partial screenshot of the assessment tool used by experts in the field to evaluate each of the annotations made by OnBaSAM.

to indicate which concepts might be missing, by creating a manual annotation. The detection of missing concepts by the experts allowed us to enrich the ontology. We used three widely used standard measurements [34], to evaluate annotation performance and the relevance of the concepts: precision, recall, and the F-measure. Five experts evaluated 410 events. The resulting scores yielded a precision of 0.91, a recall of 0.9, and an F-Measure of 0.91. The results presented may be moderate, given certain biases in the evaluation [34]. Indeed, the presence of annotations on the corpus influences the annotation of the evaluators who will focus on the present annotations and not on the non-annotated data. Based on an analysis of these results, we modified OntoPaRON and used the new version to annotate a corpus of 928 patient files. We used JMP software for statistical data processing. Data were processed by merging the sociodemographic data for the patients and the semantic annotation data into a single table.

## Results

We annotated 928 patient files from the ALS-IDF network between January 2, 2013 and December 31, 2017. These years were chosen on the basis of the start and end dates of this project (2014—May 2019). The annotation of these 928 files represented 31,260 events, or more than 1,000 000 words. The sociodemographic data for the patients are summarized in Table 4. The population consisted of 52% men and 48% women (sex ratio of 1.09). The patients had a spinal form of the disease in 65% of cases and a bulbar form in 34%. The bulbar form was more frequent in women and the spinal form was more frequent in men [35]. Mean age at diagnosis was 65.2 years, and mean age at inclusion in the ALS-IDF network was 66 years. The mean duration of support from the ALS-IDF network was 550 days. Most of the

**Table 4. Socio-demographic data of the study population.**

|  |  | Women (*n* = 442) | Men (*n* = 486) | Total (*n* = 928) |
|---|---|---|---|---|
| Age at diagnosis | mean | 67.15 | 63.4 | 65.2 |
|  | SD | 11.77 | 11.88 | 11.97 |
|  | Min-Max | 22-90 | 20-89 | 20-90 |
| Inclusion age | mean | 67.76 | 64.16 | 65.9 |
|  | SD | 11.76 | 11.96 | 12 |
|  | Min-Max | 22-91 | 20-91 | 20-91 |
| Form of the pathology | Spinal form | 247 (60%) | 319 (71%) | 566 (65%) |
|  | Bulbar form | 164 (40%) | 130 (29%) | 294 (34%) |
| Follow-up in days | mean | 548 | 552.6 | 550.3 |
|  | SD | 446.8 | 427 | 436.6 |
|  | Min-Max | 6-2302 | 6-2207 | 6-2302 |
| Number of events | mean | 34.6 | 32.6 | 33.6 |
|  | SD | 31.4 | 32.7 | 32.13 |
|  | Min-Max | 1-276 | 1-322 | 1-322 |
| Social Status | Single | 63 (16%) | 63 (13%) | 126 (14%) |
|  | Married | 215 (54%) | 337 (73%) | 552 (64%) |
|  | Divorced | 62 (16%) | 46 (10%) | 108 (13%) |
|  | Widower | 58 (15%) | 15 (3%) | 73 (8%) |
| Lifestyle | Lives alone | 107 (27%) | 59 (13%) | 166 (20%) |
|  | Lives with family | 282 (70%) | 384 (86%) | 666 (78%) |
|  | Lives in an institution | 12 (3%) | 5 (1%) | 17 (2%) |

All the socio-demographic characteristics of the study population.

patients lived in families (78%) and were married (64%) and all lived in the Ile-de-France region. Support from the ALS-IDF network resulted in a mean of 33.6 reported events per patient, but there was considerable heterogeneity, with a minimum of one event for four patients, more than 200 events for others, and a maximum of 322 events entered for one patient.

## Contribution of fully defined concepts to the identification of patient needs

We illustrated the contribution of fully defined concepts to the analysis of care trajectories by analyzing some of these concepts. The percentage of patients affected by these themes during their care pathway is shown in Table 5. Some themes were expressed differently. For example, the defined concept 'field of technical aids' related of technical aids concerns almost the entire population (93%), but the fully defined concept 'field of exhaustion' was identified as present in more than half (55%) of the patients included in the SLA-IDF network.

We sought to know how these fully defined concept were expressed according to the specific characteristics of the patients with regard to sociodemographic elements (age, social status, lifestyle, etc.) and clinical variables (form of the pathology, motor state, cognitive state). We used different statistical tests such as the linear regression model, the Student's t test in comparison and the search for correlation using the Pearson test.

The fully defined concept of 'field of technical aids', which concerns 93% of the population, mainly concerns the youngest patients ($p < 0.0001$) with a spinal form ($p = 0.020$) compared to people with a bulbar form (Student's t test showed a mean least squares difference of 31,07 for spinal form and 25,00 for bulbar form). The demands for the implementation of technical aids, increase with the time of presence in the SLA-IDF network, as well as with the presence of an altered motor state ($p < 0.0001$) but not in the case of cognitive impairment ($p = 0.67$). The Table 6 shows the importance of the ranking of each clinical variable in the expression of the fully defined concepts studied.

The occurrence of exhaustion during care was linked to the presence of signs indicating motor degradation (declining motor state) ($p < 0.0001$) or cognitive impairment ($p < 0.0001$). Marital and social status appeared to be involved in exhaustion, which was more frequently among divorcees than among married patients (Student's t test showed a mean least squares difference of 4.86 for divorcees and 3.01 for married patients). The living conditions of the patient seemed to affect the likelihood of exhaustion, which was more frequent for patients living in families (mean of 4.45) than among those living alone (mean of 2.75). The occurrence of exhaustion during patients care was correlated with the recording of events by the

**Table 5. Semantic annotation of fully defined concepts.**

| Fully defined concepts | Mean / SD | Presence $n$(%) vs Absence $n$ (%) | $p$ |
|---|---|---|---|
| Fully defined concept: 'field of technical aids' | 29 / 42.23 | 859 (93%) vs 69 (7%) | <0.0001 |
| Fully defined concept: 'field of exhaustion' | 3.67 / 6.56 | 507 (55%) vs 421 (45%) | |
| Fully defined concept: 'field of human help' | 3.57 / 4.27 | 707 (76%) vs 221 (24%) | <0.0001 |
| Fully defined concept: 'area of need and request for social help' | 3.73 / 4.53 | 722 (78%) vs 206 (22%) | <0.0001 |
| Fully defined 'concept of motor state' | 4.39 / 4.40 | 782 (84%) vs 146 (16%) | <0.0001 |
| Fully defined 'concept of cognitive state' | 0.49 / 1.41 | 174 (19%) vs 754 (81%) | <0.0001 |

Quantitative and qualitative data for the semantic annotation of certain fully defined concepts.

**Table 6. Association between fully defined concepts and clinical variables asses by linear regression model.**

| Fully defined concept | Clinical variables | Log Worth | p-value |
|---|---|---|---|
| Fully defined concept 'field of technical aids' | Motor state | 48,830 | <0.0001 |
| | Age | 8,718 | <0.0001 |
| | Form of the pathology (bulbar) | 1.680 | = 0.020 |
| | Cognitive state | 0.171 | = 0.67 |
| Fully defined concept 'field of exhaustion' | Motor state | 17.621 | <0.0001 |
| | Cognitive state | 6.310 | <0.0001 |
| | Age | 0.488 | = 0.32 |
| Fully defined concept 'field of human help' | Motor state | 33.81 | <0.0001 |
| | Age | 7.56 | <0.0001 |
| | Form of the pathology | 2.529 | = 0.002 |
| | Cognitive state | 0.996 | = 0.101 |
| Fully defined concept 'area of need and request for social help' | Motor state | 43.259 | <0.0001 |
| | Age | 12.760 | <0.001 |
| | Cognitive state | 2.823 | = 0.001 |
| | Form of pathology | 1.11 | = 0.07 |

Relative effects of fully defined concept as estimated by linear regression model. False discovery rate p-value is given for each effect using the Benjamini-Hochberg technique that adjusts for multiple tests. False discovery rate LogWorth, which is the best statistic for plotting and assessing significance, is defined as -log10 (FDR p-value).

coordinators of the SLA-IDF network ($r = 0.24$; $p < 0.0001$) and necessitated a coordination action ($r = 0.5$; $p < 0.0001$), in particular the search for an appropriate structure ($r = 0.17$; $p < 0.0001$) and the provision of a human helper ($r = 0.23$; $p < 0.0001$).

The fully defined concept of 'field of human help' concerned 76% of patients during their care trajectories. The use of human help is related to the motor state, ($p < 0.0001$) person's age ($p < 0.0001$) and the initial form of the disease ($p = 0.0021$), with people with the spinal form having a greater need for human assistance than those with the bulbar form. Living conditions also influenced the need for human help ($p = 0.0011$). Student's t test showed that people living alone had a greater need for human aid than those living in a family (mean value of 4.48 for people living alone and 3.4 for those living in families).

The fully defined concept of 'area of need and request for social help' were linked to the motor state ($p < 0.0001$) of the patient and age of the person at inclusion ($p < 0.0001$) in the ALS-IDF network. Requests and needs were more numerous for younger patients and decreased with age. They therefore increased with disease progression and time spent in the ALS network ($p < 0.0059$), but were not linked to the initial form of the disease ($p = 0.07$).

## Care pathway coordination

Within the framework of coordination, we focused particularly on two fully defined concepts, investigating the difference between 'Coordination requests received' and 'Coordination actions to match resources to needs'. The allocation of necessary resources based on need can arise in two situations: a) a clearly stated request for resources or b) detection of the need during evaluation by the coordinator. The values for the annotation of these two fully defined concepts are presented in Table 7. A comparison of the two defined concepts 'Coordination requests received' and 'Coordination actions to match resources to needs' revealed a significant difference ($p < 0.0001$). This suggests that some requests are expressed explicitly, but that

**Table 7. Semantic annotation of fully defined concepts in the coordination ontology.**

| Fully defined concepts | Mean / SD |
|---|---|
| Coordination action | 74.17 / 77.70 |
| Match resources to needs | 10.41 / 11.02 |
| Coordination requests received | 5.42 / 6.55 |

Semantic annotation of fully defined concepts for specific coordination actions. In coordination actions, the number of actions matching resources to patient needs (Match resources to needs) exceeded the number of requests received by coordinators (coordination requests received) ($p < 0.0001$).

coordinators carry out assessments and propose solutions without the request being clearly expressed. We validated this hypothesis by exploring this issue at a finer level of granularity of the OntoPaRON ontology through the extraction of annotations for the concept of 'Request to find a healthcare professional'. This concept is defined as the explicit formulation of a request for coordinators to find a healthcare professional (e.g. physiotherapist, speech therapist, GP). We investigated whether the occurrence of this request was related to the resulting coordinating action, 'Search for a healthcare professional' ($r = 0.28$; $p < 0.001$).

We transformed the quantitative data into dichotomous discrete data (0 = no demand or not seeking of a healthcare professional; 1 = request made or looking for a professional) for this analysis, to determine the total number of patients concerned by these two concepts. In 64 cases (7%) a 'Request to find a healthcare professional' occurred, whereas 'Search for a healthcare professional' occurred for 209 patients (22% of the patients). Thus, in 145 cases, the coordinators searched for a healthcare professional in the absence of a direct request from the patient.

OntoPaRON can be specifically identify the type of healthcare professional sought. The healthcare professionals most sought by coordinators are shown in Table 8. For certain patients, the search for a healthcare professional related to more than one type of professional (e.g. physiotherapist and nurse; physiotherapist and doctor). Physiotherapists were the most sought category of paramedical professionals for the management of patients with ALS.

## Discussion

Improvements in the efficiency of patient care pathway will require the prior description of these trajectories, for their analysis and the identification of ways of improving them. A top-down approach can be envisaged, based on the Information Systems Medicalization Program

**Table 8. Requests and searches for healthcare professionals.**

| Request to find a healthcare professional | 64 |
|---|---|
| **Search for a healthcare professional** | **209** |
| Searches for a physiotherapist | 162 |
| Searches for a doctor | 16 |
| Searches for a speech therapist | 18 |
| Searches for a nurse | 14 |
| Searches for psychologist | 9 |

The requests to search for a healthcare profession correspond to all the requests received by the coordinators to find a healthcare professional. The number of such requests was smaller than the number of searches for healthcare professionals actually carried out ($p < 0.001$).

(PMSI) for the medical dimension [36, 37], but the social, medico-social, and coordination dimensions are not included in this program. A bottom-up approach could also be envisaged, starting from real-life patient data. We chose to follow this second approach, using textual data from a coordination network for people living with ALS in the Ile-de-France region.

We created a modular ontology for the processing of such data, taking all aspects of the patient care pathway into account: the medical, socio-environmental, and coordination dimensions. The choice of a modular system and the creation of defined concepts made it possible to group together concepts dealing with the same theme from different ontology modules under a defined concept. The themes for the defined concepts were chosen on the basis of published data for ALS. Like Grau [23], we observed the positive aspects of modularity. However, modularity requires constant attention to the positioning of defined concepts and the management and attribution of relationships between concepts.

The annotation results revealed that the expression of needs and requests, particularly for human and social aid, were expressed differentially by the patients of the ALS-IDF network. Patients' needs vary according to clinical variables such as motor status (i.e. the motor progression of the disease), the initial form of the disease, the presence of cognitive impairment, the age and living conditions. Our quantitative approach revealed large differences between patients in the number of events recorded. It is important to take this first criterion into account in analyses of patient care trajectory, as it indicates that some patients make more requests and have a greater need for the coordination of their care (care management) than others [38]. The evolution of motor impairments is the most important factor in the care pathway. It intervenes at the same time in the setting up of technical assistance, in the appearance of caregiver exhaustion, in the setting up of human assistance as well as in social demands. The use of technical and human aid was more frequent for the spinal than for the bulbar form, consistent with the natural course of disease for this form. These results are comprehensible, given the characteristics of spinal involvement, which mostly affects the limbs, limiting the patient's ability to perform activities of daily living and necessitating technical or human aid to compensate for the disability. These needs change over time and concern all areas of daily life [39].

The age of the patient also affects the type of aid required (human and technical), needs, and social requests. In France, age is an important criterion in social and medico-social policy. It directs and defines the type of social benefits that can be claimed, according to the person's situation and disability. Such aid requires funding, with out-of-pocket expenses greater for those over 60 years of age, which may lead some to renounce such assistance [40]. Our results also linked the occurrence of exhaustion with cognitive alterations, consistent with published findings [41, 42].

The structure of the ontology made it possible to quantify the requests made to coordinators and the actions implemented. The number of requests made to coordinators was smaller than the number of actions implemented. Coordinators probably took preventive action, by analyzing and assessing certain needs that were not identified by patients or their families. The level of granularity of OntoPaRON made possible the specific identification of requests from patients to find healthcare professionals who would come to their homes, and to quantify the searches for such professionals made by coordinators. We wanted to know whether such requests to search for healthcare professionals were related to healthcare availability (density of healthcare professionals) in the local area (department, about the size of a county). A comparison of these results with data from the Regional Health Agency of Ile-de-France (https:// santegraphie.fr/accueil/accueil) for the density of healthcare professionals showed that searches for physiotherapists were not always linked to areas in which such professionals were in short supply. Such requests are probably made when patients begin to encounter increasing difficulty leaving their home to visit the physiotherapist's office. The continuation of care

therefore requires the healthcare professional to come to the patient's home. We hypothesize that such professionals may be present in a given area but that they do not provide, or refuse to provide, home care (probably due to the sum they are reimbursed for such visits). An absence or interruption of care can have a major impact on the care pathway of the patient, who may require hospitalization due to a deterioration of their medical condition or the exhaustion of their carers.

We found that alignment was better in the medical domain than, for example, in the socio-environmental domain. The granularity of our ontology, related to our original objective of annotating specific textual data and the specificities of the French social and medical sector, explain why these classes were not found in many ontologies, and fully justifies the creation of the various modules of the OntoPaRON ontology.

Our study had several limitations. First, only a limited number of patients. The ALS-IDF network provides support for a large proportion of patients with ALS in Ile-de-France, but these data may not be generalizable to all patients with ALS in France, as the monitoring and care services available differ between the regions of France. Second, we did not account for the timing or chronology of the appearance of various difficulties. The time between requests for care or assistance and their implementation can have an impact, worsening the difficulties encountered. The identification of interruptions of care, requires an identification of the criteria for avoidable and necessary hospitalizations (for respiratory decompensation, for example). It was not possible to perform such an analysis within the time constraints of our project, and further studies are required to determine whether there are predictive factors or indicators. The semantic annotation generated numerous quantitative results, but not all the data were processed, due to the short duration of this project. The exploitation and analysis of all the results should provide us with a more precise vision of the elements of care trajectories. Our results cover all the patients included, revealing specific features for some. A more detailed cluster analysis would make it possible to observe the trajectories and determine whether certain sequences (of events or difficulties) are identifiable. We have also used OntoPaRON to annotate coordination corpora for Parkinson's disease. The initial results showed that two of the modules (coordination and socio-environmental) were suitable for report annotation, but the corpus volume was too small validate the results. In addition, we have started tests to use our approach in psychiatry. The first results show that the semantic annotator works in another domain but this remains to be further investigated.

As described in section "Modularity of OntoPaRON", we organize our ontologies with a core ontology built for a long time in our laboratory, TopMenelas. It allows us (i) to have a foundational ontology to subsumerate, and (ii) to ensure consistency between the different ontologies developed. But we are aware that there are, among others, two federative initiatives, DOLCE with the core BTL2 ontology (http://biotopontology.github.io) and BFO with the core OGMS ontology (https://bioportal.bioontology.org/ontologies/OGMS). We have begun a work of alignment of TopMenelas with these two ontologies, in order to take advantage of the eco-system of these two initiatives.

In conclusion, we present one approach, based on the reasoning capacity and possible inferences in ontologies, for creating defined concepts with not only a semantic aspect but also a real dimension and clinical expression. The modularization of ontologies and their association with the automatic NLP tools, as developed here, will make it possible to annotate french corpora and to extract knowledge. Further studies of annotations will be required to identify the causes of interruptions in care. This initial work highlights the variability of needs and demands in the care pathway of individuals not only on the basis of medical criteria, such as the initial form of the disease (spinal or bulbar), but also on the basis of intrinsic criteria, such as patient age or living conditions. We are currently using and continuing to develop the

OnBaSAM annotation tool in the framework of the university hospital health research project PSY-CARE.

## Supporting information

**S1 Table. List of some of the fully defined concepts in OntoPaRON ontology.** List of some of the defined concepts and their formal definitions, present in each module of the Onto-PaRON modular ontology. We recall that the HermiT reasoner infers the membership of all concepts sharing a relationship to a fully defined concept defined with this same relationship. (PDF)

## Author Contributions

**Conceptualization:** Sonia Cardoso, Pierre Meneton, David Grabli, Jean Charlet.

**Data curation:** Sonia Cardoso.

**Formal analysis:** Sonia Cardoso, Pierre Meneton, Jean Charlet.

**Funding acquisition:** Sonia Cardoso.

**Investigation:** Sonia Cardoso, Jean Charlet.

**Methodology:** Sonia Cardoso, Gilles Guezennec, Jean Charlet.

**Project administration:** Sonia Cardoso, Vincent Meininger.

**Resources:** Sonia Cardoso.

**Software:** Gilles Guezennec.

**Writing – original draft:** Sonia Cardoso, Jean Charlet.

**Writing – review & editing:** Pierre Meneton, Xavier Aimé, Vincent Meininger, Gilles Guezennec, Jean Charlet.

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
