## [Decision Letter · Decision Letter 0]

17 Sep 2020

PONE-D-20-20582

Use of a modular ontology and a semantic annotation tool to describe the care pathway of patients with amyotrophic lateral sclerosis in a coordination network.

PLOS ONE

Dear Dr. CARDOSO,

Thank you for submitting your manuscript to PLOS ONE. After careful consideration, we feel that it has merit but does not fully meet PLOS ONE’s publication criteria as it currently stands. Therefore, we invite you to submit a revised version of the manuscript that addresses the points raised during the review process.

We look forward to receiving your revised manuscript.

Kind regards,

Robert Hoehndorf, Ph.D.

Academic Editor

PLOS ONE

Journal Requirements:

"The authors have declared that no competing interests exist"

We note that one or more of the authors are employed by a commercial company: Cogsonomy.

2.1. Please provide an amended Funding Statement declaring this commercial affiliation, as well as a statement regarding the Role of Funders in your study. If the funding organization did not play a role in the study design, data collection and analysis, decision to publish, or preparation of the manuscript and only provided financial support in the form of authors' salaries and/or research materials, please review your statements relating to the author contributions, and ensure you have specifically and accurately indicated the role(s) that these authors had in your study. You can update author roles in the Author Contributions section of the online submission form.

2.2. Please also provide an updated Competing Interests Statement declaring this commercial affiliation along with any other relevant declarations relating to employment, consultancy, patents, products in development, or marketed products, etc.  

3. Please ensure that you refer to all your Figures in your text as, if accepted, production will need this reference to link the reader to the figures.

4. We note you have included tables to which you do not refer in the text of your manuscript. Please ensure that you refer to Tables in your text; if accepted, production will need this reference to link the reader to the Table.

Reviewers' comments:

Reviewer's Responses to Questions

**Comments to the Author**

1. Is the manuscript technically sound, and do the data support the conclusions?

Reviewer #1: Yes

Reviewer #2: Partly

2. Has the statistical analysis been performed appropriately and rigorously? 

Reviewer #1: Yes

Reviewer #2: I Don't Know

3. Have the authors made all data underlying the findings in their manuscript fully available?

Reviewer #1: No

Reviewer #2: Yes

4. Is the manuscript presented in an intelligible fashion and written in standard English?

Reviewer #1: No

Reviewer #2: Yes

5. Review Comments to the Author

Reviewer #1: In this manuscript, titled ‘Use of a modular ontology and a semantic annotation tool to describe the care pathway of patients with amyotrophic lateral sclerosis in a coordination network’, the authors developed the tools, OntoPaRON and OnBaSAM, to describe the care pathway of patients with ALS based on real-life textual data from the Ile-de-France ALS network. Authors hope to identify the difficulties and needs of patients and their families at home, to understand the coordination actions implemented and to identify situations or types of patients confronted with multiple difficulties by analyzing of the textual data.

Since, the references, figures and tables were not cited correctly, not easy to well understand the manuscript and go through the tools which represented in the references and used in this study. Even though, the model and algorithms used in this study are simple and not novel but authors applied them in a good way.

Then, I’d rather recommend this manuscript to be published neither as current version nor minor-correction version. I believe that they need to re-organize the study to make a serious effort on improving the writing. In the following sections, the specific requirements for future publication of this study are explained in detail.

1) In the ‘Modularity of OntoPaRON’ section, it is mentioned that OntoPaRON has four modules and Table 2 showed four modules, while five modules were defined. Also, there is no reference or explanation about how and why these modules were chosen.

2) The concepts in each module should be listed. In addition, Fig. 2 shows the OntoPaRON inheritance diagram but there is no explanation about how the connecting arrows was drawn in this figure.

3) It is mentioned that the ontology includes 43 fully defined concepts. It is recommended that authors include the list of all fully defined concepts with their concepts as a supplementary data.

4) Authors used a linear regression model to investigate whether the identified themes specifically concerned patients with particular characteristics. First of all, the common term for assigning independent variables in regression is ‘X’ and ‘Y’ for dependent variable. So, it is better to change the terms to prevent the misunderstanding. It would be also interesting to see the feature importance based extracted from regression model to interpret about the importance of each independent variable explaining the fully defined concept.

Minor points

• All the references, figures and tables should be cited well in the entire manuscript.

• I didn’t care much of typos, punctuations and grammar mistakes but there are several mistakes which authors should ask for English proof reading to improve the writing. Also, the authors need to systematically organize the usage of acronyms. Some of them not using anymore through the manuscript after the first occurrence, knowledge engineering (KE), and some mentioned for two times, natural language processing (NLP).

• What is the parenthesis means in Table 3?

• It is recommended that the numbers with more than 3 digits separate by ‘,’ not space and using ‘.’ for decimals.

Reviewer #2: This paper describes the creation of an ontology and associated tool for characterisation and management of patients with ALS, using French textual data. Overall the study is interesting, and has produced several potentially interesting outcomes that consist in an ontology, surrounding analysis tools, and disease insights that could contribute to improved patient management.

My overall comment is that certain aspects of the methodology and results are somewhat unclear, and should be improved before publication. I am, therefore, making the suggestion of major changes, not because the content of the paper is bad (this is not the case), but because some of the argumentation and explanation needs to be reformulated and extended, and there are significant formatting problems that inhibit understanding of the paper.

--

> Our approach was unique inits creation of fully defined concepts at different levels of the modular ontology to address specific topics relating to healthcare trajectories.

I am not entirely sure what this means, there are many ontologies and associated tools that approach characterisation of different facets of disease management (for a recent example, see the COVID-19 ontology http://www.aber-owl.net/ontology/COVID-19/ ). I would suggest that the unique approach here consists more in:

- The application of ontology technology to French language clinical text

- The creation of a new ontology-based semantic annotation / analysis tool

- The use of the ontology to gain insight into the disease (this is mentioned in the latter part of the abstract)

The later methods and results discuss the idea of 'fully defined concepts' in more detail, but it's unclear to me how they differ from the ontology modules, or what exactly makes them fully defined, as opposed to other concepts or modules in the ontology.

--

The introduction discusses ALS deeply, and provides some other examples of disease-specific ontologies. However, little time is spent on applications of ontology to the area of text mining. I would suggest this review as a start: http://cobweb.cs.uga.edu/~kochut/teaching/8350/Papers/Ontologies/TextMining-RawText.pdf

--

> A modular ontology seemed be the most appropriate model for taking all these aspects into account.

It is unclear what a 'modular ontology' is in this context. Also the choice to use a modular design should be better explained. In fact, I note that this term is defined later in the paper, and the choice explained better (although the definition could perhaps be cited). Perhaps this implementation detail can be omitted entirely from the introduction, and left to the later, better explanation?

--

There is a helpful guideline on minimum reporting for ontologies. I do not suggest that any of these , but the authors could add a note https://jbiomedsem.biomedcentral.com/articles/10.1186/s13326-017-0172-7

--

I think the explanation of the OnBaSAM tool, starting on page 8, should be further developed. What is unclear to me, is how these tools implement or replace functionality provided by the GATE framework itself. The GATE framework natively supports the use of ontologies as a resource for text mining, as evidenced by the documentation: https://gate.ac.uk/sale/tao/splitch14.html , as well as the pre-processing steps mentioned.

The discussion could also include some mention of how generalisable the tools for text mining and analysis are. Could they be used with other domain ontologies? Are they available on the web?

--

The methods of the validation should be more explicit: what exactly were the expert evaluators asked to do? It's unclear whether they only verified the machine-derived labels, or also created an annotation themselves. In the former case, recall is not an informative measure (or at least, it is misleading, as it does not describe the proportion of the actually existent concepts found).

Optionally, the authors could consider measuring also inter-annotator agreement for the validation stage of their annotations. If not, they should mention potential limitations arising from treating non-perfect/human operators as a gold standard for evaluation. I can't further comment on the evaluation with more information about the methods.

--

The ontology itself is interesting, using its own defined upper-level stratification of terms, with entity, abstract object, and ideal object. I don't think this is necessarily a bad thing, however many biomedical ontologies use the Basic Formal Ontology (BFO), to the extent they make upper level metaphysical distinctions at all. This can, in some cases, help with integration of concepts between different ontologies. It may be worth adding a small discussion of why the authors chose this different method.

--

Citation number are missing in the document text, although they are listed in the references. I would suggest that 'pdflatex' should be run once more, to fill in the citation numbers in the document :-)

In addition, the links given as citation for people living with ALS in France and America in the introduction lead to 404 errors. Several other links included as citations are also broken, which is possibly another issue caused by the above problem.

--

The 'Construction of the OntoPaRON ontology' section mentions a standard format of single quotes for OntoPaRON and italic for relationships in the ontology. It would be helpful if the mention of italic font was italicised here to provide an example. Furthermore, it would aid understanding to highlight such terms in table 1

--

Some previous work that could be interesting for French language ontology is WHOFRE ( http://www.aber-owl.net/ontology/WHOFRE ). Could this work potentially be integrated?

--

In table 1, the language names English and French should be capitalised.

6. PLOS authors have the option to publish the peer review history of their article (what does this mean?). If published, this will include your full peer review and any attached files.

Reviewer #1: No

Reviewer #2: **Yes: **Luke T Slater

---

## [Author Response · Author response to Decision Letter 0]

19 Nov 2020

Journal Requirements:

RESPONSE: We have checked and attest that all formatting and style requirements have been met. 

"The authors have declared that no competing interests exist"

We note that one or more of the authors are employed by a commercial company: Cogsonomy.

2.1. Please provide an amended Funding Statement declaring this commercial affiliation, as well as a statement regarding the Role of Funders in your study. If the funding organization did not play a role in the study design, data collection and analysis, decision to publish, or preparation of the manuscript and only provided financial support in the form of authors' salaries and/or research materials, please review your statements relating to the author contributions, and ensure you have specifically and accurately indicated the role(s) that these authors had in your study. You can update author roles in the Author Contributions section of the online submission form.

RESPONSE Updated Funding statement : 

Xavier Aimé works in the company Cogsonomy that he created: This company's sole role is to finance training and other project management assistance for Xavier Aimé. Cogsonomy did not play a role in the study design, data collection and analysis, decision to publish, preparation of the manuscript and only provided financial support in the form of authors' salaries and/or research materials.

The funders (universities, hospitals) provided support in the form of salaries for authors [SC, PM, VM, DG, GG, JC], but did not have any additional role in the study design, data collection and analysis, decision to publish, or preparation of the manuscript. 

The specific roles of these authors are articulated in the ‘author contributions’ section.

2.2. Please also provide an updated Competing Interests Statement declaring this commercial affiliation along with any other relevant declarations relating to employment, consultancy, patents, products in development, or marketed products, etc. 

RESPONSE updated Funding competing interests: 

Xavier Aimé works in the company Cogsonomy that he created. This commercial affiliation (Cogsonomy) does not alter our (all the authors) adherence to PLOS ONE policies on sharing data and materials. 

3. Please ensure that you refer to all your Figures in your text as, if accepted, production will need this reference to link the reader to the figures.

RESPONSE: We have reformatted the manuscript according to the above style guidelines.

4. We note you have included tables to which you do not refer in the text of your manuscript. Please ensure that you refer to Tables in your text; if accepted, production will need this reference to link the reader to the Table.

RESPONSE: We have reformatted the manuscript according to the above style guidelines.

 

Reviewers' comments:

Reviewer's Responses to Questions

Comments to the Author

1. Is the manuscript technically sound, and do the data support the conclusions?

Reviewer #1: Yes

Reviewer #2: Partly

2. Has the statistical analysis been performed appropriately and rigorously? 

Reviewer #1: Yes

Reviewer #2: I Don't Know

3. Have the authors made all data underlying the findings in their manuscript fully available?

Reviewer #1: No

Reviewer #2: Yes

4. Is the manuscript presented in an intelligible fashion and written in standard English?

Reviewer #1: No

Reviewer #2: Yes

5. Review Comments to the Author

Reviewer #1: In this manuscript, titled ‘Use of a modular ontology and a semantic annotation tool to describe the care pathway of patients with amyotrophic lateral sclerosis in a coordination network’, the authors developed the tools, OntoPaRON and OnBaSAM, to describe the care pathway of patients with ALS based on real-life textual data from the Ile-de-France ALS network. Authors hope to identify the difficulties and needs of patients and their families at home, to understand the coordination actions implemented and to identify situations or types of patients confronted with multiple difficulties by analyzing of the textual data.

Since, the references, figures and tables were not cited correctly, not easy to well understand the manuscript and go through the tools which represented in the references and used in this study. Even though, the model and algorithms used in this study are simple and not novel but authors applied them in a good way.

Then, I’d rather recommend this manuscript to be published neither as current version nor minor-correction version. I believe that they need to re-organize the study to make a serious effort on improving the writing. In the following sections, the specific requirements for future publication of this study are explained in detail.

RESPONSE: We are grateful for the comments and assessment provided by the reviewer concerning the previous version of our manuscript. Please find below a detailed explanation on how we have attempted to address his/her comments.

1) In the ‘Modularity of OntoPaRON’ section, it is mentioned that OntoPaRON has four modules and Table 2 showed four modules, while five modules were defined. Also, there is no reference or explanation about how and why these modules were chosen.

RESPONSE: Thank you for your comment. We have added an extra column in table 2, to integrate the consolidation module, adding the specificity of this module as a legend. The number of fully defined classes, relationships and concepts present in each module of the OntoPaRON ontology. As shown in the table, the consolidation module does not model any concepts; its main function is to aggregate the four modules. 

We chose to use modularity for our ontology, for its reusability, easier management of complexity, customization and extensibility features, defined by different authors. The choice to create these modules results from the analyses made from the initial corpus. The analysis of the term candidates highlighted the fields: medical, socio-environmental and coordination. From this information we decided to create a modular ontology with the integration of each of these domains. For the final ontology to be the most optimal and to be secondarily reusable a core module was needed, as well as a consolidation module to aggregate all the modules. The final ontology of OntoPaRON results from the aggregation of the different modules and the use of the Protege's reasoning. We have modified part of the paragraph Construction of the OntoPaRON ontology, to make it easier to understand our choice of modularization.

These four dimensions oriented us towards the creation of four ontological modules for each of these domains. 

2) The concepts in each module should be listed. In addition, Fig. 2 shows the OntoPaRON inheritance diagram but there is no explanation about how the connecting arrows was drawn in this figure.

RESPONSE: Thank you for your comment. It is not possible to list all of the concepts present in the ontology modules, as they represent a total of 2,462 concepts. Table 2, shows the number of classes for each ontology module. All the concepts of the different modules are available on bioportal https://bioportal.bioontology.org/ontologies/ONTOPARON. 

To improve the understanding of how to import the modules in Figure 2, we have explained the meaning of import in the legend of the figure. “The arrows point in the direction of the modules that perform the import. Thus, the ontologies of the domain (ontoparonmed: medical ontology, ontoparonsoc: socio-environmental ontology; ontoparoncoord: coordination ontology) import the core ontology. In the same way OntoPaRON ontology imports each of the ontologies of the domain and by inference the core ontology.”

3) It is mentioned that the ontology includes 43 fully defined concepts. It is recommended that authors include the list of all fully defined concepts with their concepts as a supplementary data.

RESPONSE: As you proposed, we have placed in the supporting information section, a table S1 containing a list of most of the fully defined concepts of our ontology and their formal definitions.

S1 Table. List of some of the fully defined concepts in OntoPaRON Ontology. List of some of the defined concepts and their formal definitions, present in each module of the OntoPaRON modular ontology. We recall that the HermiT reasoner infers the membership of all concepts sharing a relationship to a fully defined concept defined with this same relationship. 

4) Authors used a linear regression model to investigate whether the identified themes specifically concerned patients with particular characteristics. First of all, the common term for assigning independent variables in regression is ‘X’ and ‘Y’ for dependent variable. So, it is better to change the terms to prevent the misunderstanding. It would be also interesting to see the feature importance based extracted from regression model to interpret about the importance of each independent variable explaining the fully defined concept.

RESPONSE: We thank the reviewer for pointing this out. We have revised the manuscript by adding Table 6 ‘Association between fully defined concepts and clinical variables assessed by linear regression model’ which shows the strength and ranking of the associations. These results underline the importance of the patient's motor state and age in the formulation of requests and needs. To bring more clarity to the reader we have reworded parts of the results paragraph to emphasize the role of these clinical variables.

Minor points

• All the references, figures and tables should be cited well in the entire manuscript.

RESPONSE: We apologize for our mistake. This revised version contains all the references.

• I didn’t care much of typos, punctuations and grammar mistakes but there are several mistakes which authors should ask for English proof reading to improve the writing. Also, the authors need to systematically organize the usage of acronyms. Some of them not using anymore through the manuscript after the first occurrence, knowledge engineering (KE), and some mentioned for two times, natural language processing (NLP).

RESPONSE: We are grateful for the comments, we’ve corrected the typo, and acronyms. We had the article proofread to improve English.

• What is the parenthesis means in Table 3? 

RESPONSE: Thank you for your comment. The parenthesis in Table 3 correspond to the percentage of common classes between ontology ontoparon and the reference terminology. In order to promote understanding, we have added a sentence in the table legend. ‘The table shows for each module of the ontology OntoPaRON the number of concepts aligned with the reference terminology, and the percentage of terms aligned in each module’.

• It is recommended that the numbers with more than 3 digits separate by ‘,’ not space and using ‘.’ for decimals.

RESPONSE: We thank the reviewer for pointing this out. We have revised. We have changed all the numeric data in the revised manuscript.

 

Reviewer #2: This paper describes the creation of an ontology and associated tool for characterisation and management of patients with ALS, using French textual data. Overall the study is interesting, and has produced several potentially interesting outcomes that consist in an ontology, surrounding analysis tools, and disease insights that could contribute to improved patient management.

My overall comment is that certain aspects of the methodology and results are somewhat unclear, and should be improved before publication. I am, therefore, making the suggestion of major changes, not because the content of the paper is bad (this is not the case), but because some of the argumentation and explanation needs to be reformulated and extended, and there are significant formatting problems that inhibit understanding of the paper.

RESPONSE: We are grateful for the comments and assessment provided by the reviewer concerning the previous version of our manuscript. Please find below a detailed explanation on how we have attempted to address his/her comments.

> Our approach was unique inits creation of fully defined concepts at different levels of the modular ontology to address specific topics relating to healthcare trajectories.

I am not entirely sure what this means, there are many ontologies and associated tools that approach characterisation of different facets of disease management (for a recent example, see the COVID-19 ontology http://www.aber-owl.net/ontology/COVID-19/ ). I would suggest that the unique approach here consists more in:

- The application of ontology technology to French language clinical text

- The creation of a new ontology-based semantic annotation / analysis tool

- The use of the ontology to gain insight into the disease (this is mentioned in the latter part of the abstract)

The later methods and results discuss the idea of 'fully defined concepts' in more detail, but it's unclear to me how they differ from the ontology modules, or what exactly makes them fully defined, as opposed to other concepts or modules in the ontology.

RESPONSE: We note in the introduction and conclusion that this is a specific treatment of French. The fact that the annotation tool is based on an ontology has been clarified. The part about fully defined concepts is developed with in particular supporting information and new descriptions.

The introduction discusses ALS deeply, and provides some other examples of disease-specific ontologies. However, little time is spent on applications of ontology to the area of text mining. I would suggest this review as a start: http://cobweb.cs.uga.edu/~kochut/teaching/8350/Papers/Ontologies/TextMining-RawText.pdf

RESPONSE: We appreciate your suggestion. We have included the tasks performed in our work namely information and data mining as reported in the proposed reference.

> A modular ontology seemed be the most appropriate model for taking all these aspects into account.

It is unclear what a 'modular ontology' is in this context. Also the choice to use a modular design should be better explained. In fact, I note that this term is defined later in the paper, and the choice explained better (although the definition could perhaps be cited). Perhaps this implementation detail can be omitted entirely from the introduction, and left to the later, better explanation?

RESPONSE: This observation is correct. We have changed the introduction, including a definition of modular ontology. We have therefore developed our own ontology, OntoPaRON, including these different dimensions in a modular structure. A modular ontology corresponds to a set of modules, where each module is a stand-alone component that maintains relationships with other ontology modules [16]. 

There is a helpful guideline on minimum reporting for ontologies. I do not suggest that any of these , but the authors could add a note https://jbiomedsem.biomedcentral.com/articles/10.1186/s13326-017-0172-7

RESPONSE: Good idea. The metadata has been enriched following the [Miro] principles of ontology to give more accurate information to users.

--

I think the explanation of the OnBaSAM tool, starting on page 8, should be further developed. What is unclear to me, is how these tools implement or replace functionality provided by the GATE framework itself. The GATE framework natively supports the use of ontologies as a resource for text mining, as evidenced by the documentation: https://gate.ac.uk/sale/tao/splitch14.html , as well as the pre-processing steps mentioned.

RESPONSE: We thank the reviewer for pointing this out. The reviewer is correct, GATE allows the use of ontologies as a resource for text mining. However, to be able to use it in our work we had to adapt the Gate modules. In fact GATE is adapted for English text data. The treatment of the French language requires adaptations of GATE, especially in the treatment of negation, but also in the spelling treatment. The revised text reads as follows on: Based on GATE which allows to build a semantic annotation string using an ontology. We have built a specific processing chain that we have adapted to our use case. Indeed, GATE is used and developed in English. Based on GATE, we have built a specific processing chain for our problem, which allows us to take into account the French language, the negation, as well as the export of annotations. 

The discussion could also include some mention of how generalisable the tools for text mining and analysis are. Could they be used with other domain ontologies? Are they available on the web?

RESPONSE: Thank you for your comment. We agree and we have developed this in the antepenultimate paragraph of the discussion.

We created a modular ontology for the processing of such data, taking all aspects of the patient care pathway into account: the medical, socio-environmental, and coordination dimensions. The choice of a modular system and the creation of defined concepts made it possible to group together concepts dealing with the same theme from different ontology modules under a defined concept. The themes for the defined concepts were chosen on the basis of published data for ALS. Like Grau [23], we observed the positive aspects of modularity. However, modularity requires constant attention to the positioning of defined concepts and the management and attribution of relationships between concepts. 

--

The methods of the validation should be more explicit: what exactly were the expert evaluators asked to do? It's unclear whether they only verified the machine-derived labels, or also created an annotation themselves. In the former case, recall is not an informative measure (or at least, it is misleading, as it does not describe the proportion of the actually existent concepts found).

Optionally, the authors could consider measuring also inter-annotator agreement for the validation stage of their annotations. If not, they should mention potential limitations arising from treating non-perfect/human operators as a gold standard for evaluation. I can't further comment on the evaluation with more information about the methods.

RESPONSE: As suggested by the reviewer, we have provided elements of understanding of the evaluation phase of the annotations made by experts in the field. The experts had the task of evaluating the annotations made by the annotator but also to create manual annotations if some concepts were not annotated. The revised text reads as follows on: During the evaluation process we asked the coordinators both to evaluate the concepts annotated by the system, and to indicate which concepts might be missing, by creating a manual annotation. The detection of missing concepts by the experts allowed us to enrich the ontology. 

--

The ontology itself is interesting, using its own defined upper-level stratification of terms, with entity, abstract object, and ideal object. I don't think this is necessarily a bad thing, however many biomedical ontologies use the Basic Formal Ontology (BFO), to the extent they make upper level metaphysical distinctions at all. This can, in some cases, help with integration of concepts between different ontologies. It may be worth adding a small discussion of why the authors chose this different method.

RESPONSE: We agree and we have developed this in the penultimate paragraph of the discussion.

--

Citation number are missing in the document text, although they are listed in the references. I would suggest that 'pdflatex' should be run once more, to fill in the citation numbers in the document :-)

RESPONSE: Thank you for pointing this out. We apologize for our mistake. This revised version contains all the references.

In addition, the links given as citation for people living with ALS in France and America in the introduction lead to 404 errors. Several other links included as citations are also broken, which is possibly another issue caused by the above problem.

RESPONSE: Thank you for pointing this out. We apologize for our mistake. We have changed the links and made the choice to integrate a reference in a bibliography, which will be more reliable. Moisan F, Kab S, Moutengou E, Boussac-Zerebska M, Carcaillon-Bentata L, Elbaz A. Fréquence de la maladie de Parkinson en France. Données nationales et régionales 2010-2015.; p. 69. 

--

The 'Construction of the OntoPaRON ontology' section mentions a standard format of single quotes for OntoPaRON and italic for relationships in the ontology. It would be helpful if the mention of italic font was italicised here to provide an example. Furthermore, it would aid understanding to highlight such terms in table 1

RESPONSE: We thank the reviewer for pointing this out. Table 1 illustrates the pseudo-anonymization process and not the semantic annotation process, which is illustrated in Figure 4. The purpose of table 1 is to illustrate with 2 examples the type of textual data we have processed during our work. The textual data contained many personal data (patient's name, professional name, place of residence) but also many abbreviations. In order to be able to process them, we set up a processing chain to transform the nominal data into a concept, which could be replayed during the annotation. Thus, it is possible to identify the interactions between agents.

We have made changes to the legend to clarify these points.

--

Some previous work that could be interesting for French language ontology is WHOFRE ( http://www.aber-owl.net/ontology/WHOFRE ). Could this work potentially be integrated?

RESPONSE: Thank you for your comment. In our context, whofre is an interesting work, but it is not a development that highlights the treatment of French. It is more important for us to highlight the Alzheimer's ontology of Dramé, Diallo and Co. This is what we do now in the introduction (reference 13).

In table 1, the language names English and French should be capitalised.

RESPONSE: We thank the reviewer for pointing this out. We have revised.

6. PLOS authors have the option to publish the peer review history of their article (what does this mean?). If published, this will include your full peer review and any attached files.

Do you want your identity to be public for this peer review? For information about this choice, including consent withdrawal, please see our Privacy Policy.

Reviewer #1: No

Reviewer #2: Yes: Luke T Slater

---

## [Decision Letter · Decision Letter 1]

14 Dec 2020

Use of a modular ontology and a semantic annotation tool to describe the care pathway of patients with amyotrophic lateral sclerosis in a coordination network.

PONE-D-20-20582R1

Dear Dr. CARDOSO,

We’re pleased to inform you that your manuscript has been judged scientifically suitable for publication and will be formally accepted for publication once it meets all outstanding technical requirements.

Kind regards,

Robert Hoehndorf, Ph.D.

Academic Editor

PLOS ONE

Additional Editor Comments (optional):

Reviewers' comments:

Reviewer's Responses to Questions

**Comments to the Author**

1. If the authors have adequately addressed your comments raised in a previous round of review and you feel that this manuscript is now acceptable for publication, you may indicate that here to bypass the “Comments to the Author” section, enter your conflict of interest statement in the “Confidential to Editor” section, and submit your "Accept" recommendation.

Reviewer #1: All comments have been addressed

2. Is the manuscript technically sound, and do the data support the conclusions?

Reviewer #1: (No Response)

3. Has the statistical analysis been performed appropriately and rigorously? 

Reviewer #1: (No Response)

4. Have the authors made all data underlying the findings in their manuscript fully available?

Reviewer #1: (No Response)

5. Is the manuscript presented in an intelligible fashion and written in standard English?

Reviewer #1: (No Response)

6. Review Comments to the Author

Reviewer #1: (No Response)

7. PLOS authors have the option to publish the peer review history of their article (what does this mean?). If published, this will include your full peer review and any attached files.

Reviewer #1: No

---

## [Editor Report · Acceptance letter]

21 Dec 2020

PONE-D-20-20582R1 

Use of a modular ontology and a semantic annotation tool to describe the care pathway of patients with amyotrophic lateral sclerosis in a coordination network. 

Dear Dr. Cardoso:

I'm pleased to inform you that your manuscript has been deemed suitable for publication in PLOS ONE. Congratulations! Your manuscript is now with our production department. 

Kind regards, 

on behalf of

Dr. Robert Hoehndorf 

Academic Editor

PLOS ONE